# Counterfactual Learning under Rank Preservation

## Abstract

Counterfactual inference aims to estimate the counterfactual outcome given knowledge of an observed treatment and the factual outcome, with broad applications in fields such as epidemiology, econometrics, and management science. In this paper, we propose a principled approach for identifying and estimating the counterfactual outcome. Specifically, we introduce a simple and intuitive rank preservation assumption to identify the counterfactual outcome without relying on a known structural causal model. Building on this, we propose a novel ideal loss for theoretically unbiased learning of the counterfactual outcome and further develop a kernel-based estimator for its empirical estimation. Our theoretical analysis shows that the proposed ideal loss is convex, and the proposed estimator is unbiased. Extensive semi-synthetic and real-world experiments are conducted to demonstrate the effectiveness of the proposed method.

## 1 Introduction

Understanding causal relationships is a fundamental goal across various domains, such as epidemiology (Hernán & Robins, 2020), econometrics (Imbens & Rubin, 2015), and management science (Kallus & Uehara, 2020). Pearl & Mackenzie (2018) define the three-layer causal hierarchy—association, intervention, and counterfactuals—to distinguish three types of queries with increasing complexity and difficulty (Bareinboim et al., 2022). Counterfactual inference, the most challenging level, aims to explore the impact of a treatment on an outcome given knowledge about a different observed treatment and the factual outcome. For example, given a patient who has not taken medication before and now suffers from a headache, we want to know whether the headache would have occurred if the patient had taken the medication initially. Answering such counterfactual queries can provide valuable instructions in scenarios such as credit assignment (Mesnard et al., 2021), root-causal analysis (Budhathoki et al., 2022), and fair decision-making (Imai & Jiang, 2023).

Different from interventional queries, which are prospective and estimate the counterfactual outcome in a hypothetical world via only the observations obtained before treatment (as pre-treatment variables), counterfactual inference is retrospective and further incorporates the factual outcome (as a post-treatment variable) in the observed world. This inherent conflict between the hypothetical and the observed world poses a unique challenge and makes the counterfactual outcome generally unidentifiable, even in randomized controlled experiments (RCTs) (Pearl et al., 2016; Ibeling & Icard, 2020; Bareinboim et al., 2022).

For counterfactual inference, Pearl et al. (2016) proposed a three-step procedure (abduction, action, and prediction) to estimate counterfactual outcomes. However, it relies on the availability of structural causal models (SCMs) that fully describe the data-generating process (Brouwer, 2022; Xie et al., 2023). In real-world applications, the ground-truth SCM is likely to be unknown, and estimating it requires additional assumptions to ensure identifiability, such as linearity (Shimizu et al., 2006) and additive noise (Hoyer et al., 2008; Peters et al., 2014). Unfortunately, these assumptions are hard to satisfy in practice and restrict the applicability.

To tackle the above problems, various counterfactual learning approaches have been proposed with respect to different identifiability assumptions. For example, Lu et al. (2020), Nasr-Esfahany et al. (2023), and Xie et al. (2023) established the identifiability of counterfactual outcomes based on homogeneity and strict monotonicity assumptions. The homogeneity assumption posits that the exogenous variable for each individual remains constant across different interventional environments,

and the strict monotonicity assumption asserts that the outcome is a strictly monotone function of the exogenous variable given the features. In terms of counterfactual learning, Lu et al. (2020) and Nasr-Esfahany et al. (2023) adopted Pearl's three-step procedure that needs to estimate the SCM initially. In addition, Xie et al. (2023) proposed using quantile regression to estimate counterfactual outcomes that effectively avoid the estimation of SCMs. Nevertheless, it relies on a stringent assumption that the conditional quantile functions for different counterfactual outcomes come from the same model and it requires estimating a different quantile value for each individual, leading to a challenging bi-level optimization problem.

In this work, we propose a principled counterfactual learning approach with *intuitive identifiability assumptions and theoretically guaranteed estimation methods*. On one hand, for identifiability assumptions, we introduce the simple and intuitive rank preservation assumption, positing that an individual's factual and counterfactual outcomes have the same rank in the corresponding distributions of factual and counterfactual outcomes for all individuals. We prove the identifiability of counterfactual outcomes under the rank preservation assumption.

On the other hand, we further propose a theoretically guaranteed method for unbiased estimation of counterfactual outcomes. The proposed estimation method enjoys several desirable merits. First, unlike Pearl's three-step procedure, it does not necessitate a prior estimation of SCMs and thus relies on fewer assumptions than the methods proposed by Lu et al. (2020) and Nasr-Esfahany et al. (2023). Second, in contrast to the quantile regression method proposed by Xie et al. (2023), our approach neither restricts conditional quantile functions for different counterfactual outcomes to originate from the same model, nor requires estimating a different quantile value for each individual. Third, we enhance the previous learning approaches to adopt a convex loss for estimating counterfactual outcomes, which leads to a unique solution.

In summary, the main contributions are as follows: (1) We introduce the intuitive rank preservation assumption to identify the counterfactual outcomes with unknown SCM, and establish its relationship with previous homogeneity and strict monotonicity assumptions; (2) We propose a novel ideal loss for unbiased learning of the counterfactual outcome and further develop a kernel-based estimator for the ideal loss. In addition, we theoretically show that the proposed ideal loss is convex, and the proposed kernel-based estimator is consistent; (3) We conduct extensive experiments on both semi-synthetic and real-world datasets to demonstrate the effectiveness of the proposed method.

## 2 PRELIMINARIES AND PROBLEM FORMULATION

Throughout, capital letters represent random variables and lowercase letters denote their realizations.

**Structural Causal Model (SCM, Pearl, 2009).** An SCM $\mathcal{M}$ consists of a causal graph $\mathcal{G}$ and a set of structure equation models $\mathcal{F} = \{f_1, ..., f_p\}$. The nodes in $\mathcal{G}$ are divided into two categories: (a) exogenous variables $\mathbf{U} = (U_1, ..., U_p)$, which represent the environment during data generation, assumed to be mutually independent; (b) endogenous variables $\mathbf{V} = \{V_1, ..., V_p\}$, which denote the relevant features that we need to model in a question of interest. For variable $V_j$, its value is determined by a structure equation $V_j = f_j(PA_j, U_j)$, $j = 1, ..., p$, where $PA_j$ stands for the set of parents of $V_j$. SCM provides a formal language for describing how the variables interact and how the resulting distribution would change in response to certain interventions. Based on SCM, we introduce the counterfactual inference problem in the following.

**Counterfactual Inference (Pearl, 2009).** Suppose that we have three sets of variables denoted by $X, Y, \mathbf{E} \subseteq \mathbf{V}$, counterfactual inference revolves around the question, "given evidence $\mathbf{E} = \mathbf{e}$, what would have happened if we had set $X$ to a different value $x'$?". Pearl et al. (2016) propose using the three-step procedure to answer the problem: (a) **Abduction**: determine the value of $\mathbf{U}$ according to the evidence $\mathbf{E} = \mathbf{e}$; (b) **Action**: modify the model $\mathcal{M}$ by removing the structural equations for $X$ and replacing them with $X = x'$, yielding the modified model $\mathcal{M}_{x'}$; (c) **Prediction**: Use $\mathcal{M}_{x'}$ and the value of $\mathbf{U}$ to calculate the counterfactual outcome of $Y$. In this paper, we focus on estimating the counterfactual outcome for each individual. To illustrate the main ideas, we formulate the common counterfactual inference problem within the context of the backdoor criterion.

**Problem Formulation.** Let $\mathbf{V} = (Z, X, Y)$, where $X$ causes $Y$, $Z$ affects both $X$ and $Y$, and the structure equation of $Y$ is given as

$$Y = f_Y(X, Z, U_X). \tag{1}$$

Let $Y_{x'}$ denotes the potential outcome if we had set $X = x'$. The counterfactual question, "given evidence $(X = x, Z = z, Y = y)$ of an individual, what would have happened had we set $X = x'$ for this individual", is formally expressed as estimating $y_{x'}$, the realization of $Y_{x'}$ for the individual. Here, we adhere to the deterministic viewpoint of Pearl (2009) and Pearl et al. (2016), treating the value of $Y_{x'}$ for each individual as a fixed constant. According to Pearl's three-step procedure, given the evidence $(X = x, Z = z, Y = y)$ for an individual, the identifiability of its counterfactual value $y_{x'}$ can be achieved by determining the structural equation $f_Y$ and the value of $U_X$ for this individual. This is the key idea underlying most of the existing methods.

For clarity, we use $y_{x'}$ to denote the realization of the counterfactual outcome $Y_{x'}$ *for a specific individual* with observed evidence $(X = x, Z = z, Y = y)$.

## 3 ANALYSIS OF EXISTING METHODS

In this section, we elucidate the challenges of counterfactual inference. This clarification helps further analysis of current approaches. Subsequently, we summarize the existing methods, shedding light on their limitations and thereby motivating the proposal of our method.

### 3.1 CHALLENGES IN COUNTERFACTUAL INFERENCE

The main challenge lies in that the counterfactual value $y_{x'}$ is generally not identifiable, even in randomized controlled experiments (RCTs).

By definition, $y_{x'}$ is a quantity involving two "different worlds" at the same time: the observed world with $(X = x, Z = z, Y = y)$ and the hypothetical world where $X = x'$. We only observe the factual outcome $Y_x = y$ but never observe the counterfactual outcome $Y_{x'}$, which is the fundamental problem in causal inference (Holland, 1986; Morgan & Winship, 2015). This inherent conflict prevents us from simplifying the expression of $y_{x'}$ to a do-calculus expression, making it generally unidentifiable, even in RCTs (Pearl et al., 2016). Therefore, in addition to the widely used assumptions such as conditional exchangeability, overlapping, and consistency (Hernán & Robins, 2020), counterfactual inference requires extra assumptions to ensure identifiability. Essentially, estimating $y_{x'}$ is equivalent to estimating the individual treatment effect $y_{x'} - y_x$, while the conditional average treatment effect (CATE) $\mathbb{E}[Y_{x'} - Y_x | Z = z]$ represents the ATE for a subpopulation with $Z = z$, overlooking the inherent heterogeneity in this subpopulation caused by the noise terms such as $U_X$ (Albert et al., 2005; Lei & Candès, 2021; Ben-Michael et al., 2022; Jin et al., 2023).

### 3.2 SUMMARY OF EXISTING METHODS

We summarize the existing methods in terms of identifiability assumptions and estimation strategies.

We first present an equivalent expression of Eq. (1) by using the notation of $(Y_x, Y_{x'})$. Eq. (1) be reformulated as the following system

$$Y_x = f_Y(x, Z, U_x), \ Y_{x'} = f_Y(x', Z, U_{x'}),$$

where $U_x$ and $U_{x'}$ denote the values of $U_X$ given $X = x$ and $X = x'$, respectively. The exogenous variable $U_X$ denotes the background and environment information induced by many unmeasured factors (Pearl et al., 2016), and thus $U_x$ and $U_{x'}$ account for the heterogeneity of $Y_x$ and $Y_{x'}$ in the observed and hypothetical worlds, respectively. These two worlds may exhibit different levels of noise due to unmeasured factors (Heckman et al., 1997; Chernozhukov & Hansen, 2005).

For identification, previous work relies on the key homogeneity and strict monotonicity assumptions.

**Assumption 3.1** (Homogeneity). $U_x = U_{x'}$.

**Assumption 3.2** (Strict Monotonicity). *For any given $x$ and $z$, $f_Y(x, z, U_x)$ is a smooth and strictly monotonic function of $U_x$; or $Y_x = f_Y(x, z, U_x)$ is a bijective mapping from $U_x$ to $Y_x$.*

Assumption 3.1 implies that the value of $U_X$ for each individual remains unchanged across $x$. Assumption 3.2 implies that $Y_x$ is a strict monotonic function of $U_x$ in the subpopulation of $(X = x, Z = z)$. In Assumption 3.2, the smoothness and strict monotonicity of $f_Y(x, z, U_x)$ are akin to a bijective mapping of $Y_x$ and $U_x$ and serve the same purpose, so we don't distinguish

them in detail. The identifiability of $y_{x'}$ in Lu et al. (2020), Xie et al. (2023) and Nasr-Esfahany et al. (2023) depends on Assumptions 3.1-3.2, as summarized in Lemma 3.3.

**Lemma 3.3.** *Under Assumptions 3.1-3.2, $y_{x'}$ is identifiable.*

For estimation of $y_{x'}$, following Pearl's three-step procedure, Lu et al. (2020) and Nasr-Esfahany et al. (2023) initially estimate $f_Y$ and $U_X$ for each individual. However, estimating $f_Y$ and $U_X$ needs to impose extra assumptions, such as linearity (Shimizu et al., 2006) and additive noise Peters et al. (2014). On the other hand, Xie et al. (2023) demonstrate that $y_{x'}$ corresponds to the $\tau^*$-th quantile of the distribution $\mathbb{P}(Y|X = x', Z = z)$, where $\tau^*$ is the quantile of $y$ in $\mathbb{P}(Y|X = x, Z = z)$ (See the proof of Lemma 3.3 for more details). Based on it, the authors uses quantile regression to estimate $y_{x'}$, which avoids the problem of estimating $f_Y$ and $U_X$. Nevertheless, this method fits a single model to obtain the conditional quantile functions for both the counterfactual and factual outcomes. Thus, its validity relies on the underlying assumption that the conditional quantile functions of outcomes for different treatment groups stem from the same model. In addition, it involves estimating a distinct quantile value for each individual before deriving the counterfactual outcomes, posing a challenging bi-level optimization problem.

## 4 IDENTIFICATION THROUGH RANK PERSERVATION

In this section, we introduce a intuitive rank preservation assumption for identifying $y_{x'}$. *From a high-level perspective, identifying $y_{x'}$ essentially involves establishing the relationship between $Y_x$ and $Y_{x'}$ for each individual.* Pearl's three-step procedure achieves this by estimating $f_Y$ and $U_X$.

### 4.1 RANK PERSERVATION ASSUMPTION

Our identifiability assumption is based on Kendall's rank correlation coefficient defined below.

**Definition 4.1** (Kendall, 1938). *Let $(x_1, y_1), ..., (x_n, y_n)$ be a set of observations of two random variables $(X, Y)$, such that all the values of $x_i$ and $y_i$ are unique (ties are neglected for simplicity). Any pair of $(x_i, y_i)$ and $(x_j, y_j)$, if $(x_j - x_i)(y_j - y_i) > 0$, they are said to be concordant; otherwise they are discordant. The* sample *Kendall rank correlation coefficient is defined as*

$$\rho_n(X, Y) = \frac{2}{n(n-1)} \sum_{1 \le i < j \le n} sign((x_i - x_j)(y_i - y_j)),$$

*where $sign(t) = -1, 0, 1$ for $t < 0$, $t = 0$, $t > 0$, respectively. For any two random variables $(X, Y)$, we define $\rho(X, Y) = 1$, if $\rho_n(X, Y) = 1$ for all integers $n \ge 2$.*

The $\rho_n(X, Y)$ also can be written as $2(N_c - N_d)/n(n-1)$, where $N_c$ is the number of concordant pairs, $N_d$ is the number of discordant pairs. It is easy to see that $-1 \le \rho_n(X, Y) \le 1$ and if the agreement between the two rankings is perfect (i.e., perfect concordance), $\rho_n(X, Y) = 1$.

**Assumption 4.2** (Rank Preservation). $\rho(Y_x, Y_{x'}|Z) = 1$.

For the individual with observation $(X = x, Z = z, Y = y)$, we denote $(y_x = y, y_{x'})$ as its true values of $(Y_x, Y_{x'})$. Assumption 4.2 implies that for this individual, its rankings of $y_x$ and $y_{x'}$ are the same in the distributions of $\mathbb{P}(Y_x|Z = z)$ and $\mathbb{P}(Y_{x'}|Z = z)$, respectively. Therefore, we have

$$\mathbb{P}(Y_x \le y_x|Z = z) = \mathbb{P}(Y_{x'} \le y_{x'}|Z = z). \tag{2}$$

Since $y_x = y$ is observed and the distributions $\mathbb{P}(Y_x|Z = z)$ and $\mathbb{P}(Y_{x'}|Z = z)$ can be identified as $\mathbb{P}(Y|X = x, Z = z)$ and $\mathbb{P}(Y|X = x', Z = z)$, respectively, by the backdoor criterion (i.e., $(Y_x, Y_{x'}) \perp\!\!\!\perp X|Z$). Therefore, we have the following Proposition 4.3 (see Appendix A for proofs).

**Proposition 4.3.** *Under Assumption 4.2, $y_{x'}$ is identified as the $\tau^*$-th quantile of $\mathbb{P}(Y|X = x', Z = z)$, where $\tau^*$ is the quantile of $y$ in the distribution of $\mathbb{P}(Y|X = x, Z = z)$.*

Proposition 4.3 shows that Assumption 4.2 can serve as a substitute for Assumptions 3.1-3.2 in identifying $y_{x'}$. Unlike Assumptions 3.1-3.2, Assumption 4.2 is simple and intuitive, as it directly links $Y_x$ and $Y_{x'}$ for each individual. To clarify the relationship between Assumption 4.2 introduced by this work and Assumptions 3.1-3.2 from previous work, we present Proposition 4.4 below.

**Proposition 4.4.** *Under Assumption 3.1, or more generally, if $U_x$ is a strictly monotone increasing function of $U_{x'}$, Assumption 4.2 is equivalent to Assumption 3.2.*

Proposition 4.4 (see Appendix A for proofs) indicates that Assumptions 4.2 is equivalent to Assumption 3.2 under more general conditions than those considered in previous work. That is, Assumption 4.2 is slightly weaker than Assumptions 3.1-3.2 by allowing $U_{x'} \neq U_x$. For illustration, consider a SCM with $X \in \{0, 1\}$, $Y_1 = Z + U_1$, $Y_0 = Z/2 + U_0$, $U_1 = U_0^3$. In this case, $\rho(Y_0, Y_1|Z) = 1$, $U_1$ is a strictly monotone increasing function of $U_0$, but $U_1 \neq U_0$.

### 4.2 Further Relaxation of Strict Monotonicity

In Definition 4.1, we ignore ties for simplicity. However, when the outcome $Y$ is discrete or continuous variables with tied observations, $\rho(Y_x, Y_{x'})$ will always be less than 1. To accommodate such cases, we introduce a modified version of the Kendall rank correlation coefficient given below.

**Definition 4.5** (Kendall, 1945). *Let $(x_1, y_1), ..., (x_n, y_n)$ be the observations of two random variables $(X, Y)$, the modified Kendall rank correlation coefficient is define as*

$$\tilde{\rho}_n(X, Y) = \sum_{1 \leq i < j \leq n} \frac{sign((x_i - x_j)(y_i - y_j))}{\sqrt{n(n-1)/2 - T_x} \cdot \sqrt{n(n-1)/2 - T_y}},$$

*where $T_x$ is the number of tied pairs in $\{x_1, ..., x_n\}$ and $T_y$ is the number of tied pairs in $\{y_1, ..., y_n\}$. We define $\tilde{\rho}(X, Y) = 1$, if $\tilde{\rho}_n(X, Y) = 1$ for all integers $n \geq 2$.*

By comparison of Definition 4.5 and Definition 4.1, one can see that $\tilde{\rho}(X, Y)$ adjusts $\rho(X, Y)$ by eliminating the ties in the denominator, and $\tilde{\rho}(X, Y)$ reduces to $\rho(X, Y)$ if there are no ties.

**Assumption 4.6** (Rank Preservation). $\tilde{\rho}(Y_x, Y_{x'}|Z) = 1$.

Assumption 4.6 is less restrictive than Assumption 4.2 as it accommodates broader data types of $Y$. To illustrate, consider a dataset with four individuals where the true values of $(Y_x, Y_{x'})$ are $(1, 1), (2, 1.5), (2, 1.5), (3, 2.5)$. In this scenario, $\sum_{1 \leq i < j \leq n} sign((y_{i,x} - y_{j,x})(y_{i,x'} - y_{j,x'})) = 5$, $T_{Y_x} = 1$, $T_{Y_{x'}} = 1$, resulting in $\rho(Y_x, Y_{x'}) = 5/6$ and $\tilde{\rho}(Y_x, Y_{x'}) = 5/(\sqrt{6-1} \cdot \sqrt{6-1}) = 1$.

In addition, Assumption 4.6 also guarantees the identifiability of $y_{x'}$, as shown in Proposition 4.7.

**Proposition 4.7.** *Under Assumption 4.6, the conclusion in Proposition 4.3 also holds.*

## 5 Counterfactual Learning

In this section, we propose a novel estimation method for counterfactual inference. Suppose that $\{(x_i, z_i, y_i) : i = 1, ..., N\}$ is a sample consisting of $N$ realizations of random variables $(X, Z, Y)$. For an individual, given its evidence $(X = x, Z = z, Y = y)$, we aim to estimate its counterfactual outcome $y_{x'}$, which is the realization of $Y_{x'}$ for this individual.

### 5.1 The Rationale and Limitations of Using Quantile Regression

For estimating $y_{x'}$, Xie et al. (2023) formulate it as the following bi-level optimization problem

$$\tau^* = \arg\min_\tau |f_\tau(x, z) - y|, \quad f_\tau^* = \arg\min_f \frac{1}{N} \sum_{k=1}^N l_\tau(y_k - f(x_k, z_k)),$$

where $l_\tau(\xi) = \tau\xi \cdot \mathbb{I}(\xi \geq 0) + (\tau - 1)\xi \cdot \mathbb{I}(\xi < 0)$ is the check function (Koenker & Bassett, 1978), the upper level optimization is to estimate $\tau^*$, the quantile of $y$ in the distribution $\mathbb{P}(Y|X = x, Z = z)$, and the lower level optimization is to estimate the conditional quantile function $q(x, z; \tau) \triangleq \inf_y \{y : \mathbb{P}(Y \leq y|X = x, Z = z) \geq \tau\}$ for a given $\tau$. Then $y_{x'}$ can be estimated using $q(x', z; \tau^*)$.

We define the conditional quantile regression functions for $Y_x$ and $Y_{x'}$ as follows,

$$q_x(z; \tau) \triangleq \inf_y \{y : \mathbb{P}(Y_x \leq y|Z = z) \geq \tau\}, \quad q_{x'}(z; \tau) \triangleq \inf_y \{y : \mathbb{P}(Y_{x'} \leq y|Z = z) \geq \tau\}.$$

By Eq. (2), $y_{x'}$ can be expressed as $q_{x'}(z; \tau^*)$ with $\tau^*$ being the quantile of $y$ in the distribution of $\mathbb{P}(Y_x|Z = z)$, i.e., $\mathbb{P}(Y_x \leq y|Z = z) = \tau^*$. The Proposition 5.1 (see Appendix B for proofs) shows the rationale behind employing the check function as the loss for estimating conditional quantiles.

**Proposition 5.1.** *We have that (i) $q_x(Z; \tau) = \arg\min_f \mathbb{E}[l_\tau(Y_x - f(Z))]$ for any given $x$; (ii) $q(X, Z; \tau) = \arg\min_f \mathbb{E}[l_\tau(Y - f(X, Z))]$.*

There are two major concerns with the estimation method of Xie et al. (2023). First, it only fits a single quantile regression model for $q(X, Z; \tau)$ to obtain estimates of $q_x(Z; \tau)$ and $q_{x'}(Z; \tau)$. When the two conditional quantile functions $q_x(Z; \tau)$ and $q_{x'}(Z; \tau)$ originate from different models, this method may yield inaccurate estimates. Second, it explicitly requires estimating the quantile $\tau^*$ for each individual before estimating the counterfactual outcome $y_{x'}$.

Inspired by Firpo (2007), a simple improvement is to estimate $q_x(z; \tau)$ and $q_{x'}(z; \tau)$ separately. For example, for estimating $q_x(z; \tau)$, the associated loss function is given as

$$R_x(f, \tau) = \frac{1}{N} \sum_{k=1}^{N} \frac{\mathbb{I}(x_k = x) \cdot l_\tau(y_k - f(z_k))}{p_x(z_k)}, \; p_x(z) = \mathbb{P}(X = x | Z = z) \text{ is the propensity score.}$$

Likewise, we could define $R_{x'}(f, \tau)$ by replacing $x$ with $x'$. Then the estimation procedure for $y_{x'}$ involves four steps: (1) estimating the propensity score; (2) estimating $q_x(z; \tau)$ by minimizing $R_x(f, \tau)$ for a range of candidate values of $\tau$; (3) identifying the $\tau^*$ in the candidate set of $\tau$, that corresponds to the quantile of $y$ in the distribution $\mathbb{P}(Y | X = x, Z = z)$; (4) estimating $y_{x'}$ using $q_{x'}(z; \tau^*)$, where $q_{x'}(z; \tau^*)$ is obtained by minimizing $R_{x'}(f, \tau^*)$.

Despite this four-step estimation method allowing $q_x(Z; \tau)$ and $q_{x'}(Z; \tau)$ to come from different models, it still needs to estimate a different $\tau^*$ for each individual and is cumbersome.

## 5.2 Enhanced Counterfactual Learning Method

To address the aforementioned limitations of directly applying quantile regression and improve the estimation accuracy, we propose a novel loss function that yields an unbiased estimator of $y_{x'}$ for the individual with evidence $(X = x, Z = z, Y = y)$. The proposed ideal loss is constructed as

$$R_{x'}(t | x, z, y) = \mathbb{E}\left[|Y_{x'} - t| \mid Z = z\right] + \mathbb{E}\left[\text{sign}(Y_x - y) \mid Z = z\right] \cdot t,$$

which is a function of $t$ and the expectation operator is taken on the random variable of $(Y_x, Y_{x'})$ given $Z = z$. The proposed estimation method is based on Theorem 5.2.

**Theorem 5.2** (Validity of the Proposed Ideal Loss)**.** *The loss $R_{x'}(t | x, z, y)$ is convex with respect to $t$ and is minimized uniquely at $t^*$, where $t^*$ is the solution of $\mathbb{P}(Y_{x'} \leq t^* | Z = z) = \mathbb{P}(Y_x \leq y | Z = z)$.*

Theorem 5.2 (see Appendix B for proofs) implies that given the evidence $(X = x, Z = z, Y = y)$ for an individual, the counterfactual outcome $y_{x'}$ (a realization of $Y_{x'}$ for this individual) satisfies $y_{x'} = \arg\min_t R_{x'}(t | x, z, y)$ under Assumption 4.6. **Importantly**, *the loss $R_{x'}(t | x, z, y)$ neither estimates the SCM a priori, nor restricts $q_x(z; \tau)$ and $q_{x'}(z; \tau)$ stem from the same model, and it does not need to estimate a different quantile value for each individual explicitly.*

To optimize the ideal loss $R_{x'}(t; x, z, y)$, we first need to estimate it, which presents two significant challenges: (1) $R_{x'}(t | x, z, y)$ involves both $Y_x$ and $Y_{x'}$, but for each unit, we only observe one of them; (2) The terms $\mathbb{E}\left[|Y_{x'} - t| \mid Z = z\right]$ and $\mathbb{E}\left[\text{sign}(Y_x - y) \mid Z = z\right]$ in $R_{x'}(t | x, z, y)$ is conditioned on $Z = z$, and when $Z$ is a continuous variable with infinite possible values, it cannot be estimated by simply splitting the data based on $Z$. We employ inverse propensity score and kernel smoothing techniques to overcome these two challenges. Specifically, we propose a kernel-smoothing-based estimator for the ideal loss, which is given as

$$\hat{R}_{x'}(t | x, z, y) = \frac{\sum_{k=1}^{N} K_h(z_k - z) \frac{\mathbb{I}(x_k = x')}{p_{x'}(z_k)} |y_k - t|}{\sum_{k=1}^{N} K_h(z_k - z)} + \frac{\sum_{k=1}^{N} K_h(z_k - z) \frac{\mathbb{I}(x_k = x)}{p_x(z_k)} \cdot \text{sign}(y_k - y)}{\sum_{k=1}^{N} K_h(z_k - z)} \cdot t,$$

where $h$ is a bandwidth/smoothing parameter, $K_h(u) = K(u/h)/h$, and $K(\cdot)$ is a symmetric kernel function (Fan & Gijbels, 1996; Li & Racine, 2007) that satisfies $\int K(u) du = 1$ and $\int u K(u) dt = 1$, such as Epanechnikov kernel $K(u) = 3(1 - u^2) \cdot \mathbb{I}(|u| \leq 1)/4$ and Gaussian kernel $K(u) = \exp(-u^2/2)/\sqrt{2\pi}$ for $u \in \mathbb{R}$. Then we can estimate $y_{x'}$ by minimizing $\hat{R}_{x'}(t; x, z, y)$ directly.

**Proposition 5.3.** *If $h \to 0$ as $N \to \infty$, and the density function of $Z$ is twice differentiable, then*

$$\hat{R}_{x'}(t | x, z, y) \xrightarrow{\mathbb{P}} R_{x'}(t | x, z, y),$$

*where $\xrightarrow{\mathbb{P}}$ means convergence in probability.*

Table 1: $\sqrt{\epsilon_{\text{PEHE}}}$ of individual treatment effect estimation on the simulated Sim-$m$ dataset, where $m$ is the dimension of $Z$.

| Methods | Sim-5 | | Sim-10 | | Sim-20 | | Sim-40 | |
|---|---|---|---|---|---|---|---|---|
| | In-sample | Out-sample | In-sample | Out-sample | In-sample | Out-sample | In-sample | Out-sample |
| T-learner | $2.95 \pm 0.02$ | $2.66 \pm 0.01$ | $2.99 \pm 0.01$ | $3.17 \pm 0.01$ | $3.36 \pm 0.02$ | $3.19 \pm 0.03$ | $5.12 \pm 0.02$ | $4.74 \pm 0.04$ |
| X-learner | $2.94 \pm 0.01$ | $2.66 \pm 0.01$ | $2.98 \pm 0.02$ | $3.19 \pm 0.02$ | $3.31 \pm 0.02$ | $3.21 \pm 0.02$ | $5.08 \pm 0.04$ | $4.77 \pm 0.03$ |
| BNN | $2.91 \pm 0.08$ | $2.64 \pm 0.07$ | $2.90 \pm 0.11$ | $3.08 \pm 0.12$ | $3.21 \pm 0.13$ | $3.13 \pm 0.16$ | $4.81 \pm 0.10$ | $4.54 \pm 0.09$ |
| TARNet | $2.89 \pm 0.07$ | $2.64 \pm 0.06$ | $2.94 \pm 0.07$ | $3.16 \pm 0.08$ | $3.18 \pm 0.07$ | $3.11 \pm 0.07$ | $4.82 \pm 0.07$ | $4.56 \pm 0.07$ |
| CFRNet | $2.88 \pm 0.07$ | $2.62 \pm 0.06$ | $2.94 \pm 0.07$ | $3.15 \pm 0.08$ | $3.15 \pm 0.07$ | $3.08 \pm 0.07$ | $4.71 \pm 0.12$ | $4.45 \pm 0.11$ |
| CEVAE | $2.92 \pm 0.27$ | $2.65 \pm 0.21$ | $3.04 \pm 0.27$ | $3.11 \pm 0.18$ | $3.16 \pm 0.17$ | $3.11 \pm 0.17$ | $4.88 \pm 0.23$ | $4.53 \pm 0.20$ |
| DragonNet | $2.90 \pm 0.08$ | $2.63 \pm 0.08$ | $3.02 \pm 0.07$ | $3.25 \pm 0.08$ | $3.16 \pm 0.11$ | $3.09 \pm 0.10$ | $4.78 \pm 0.11$ | $4.50 \pm 0.12$ |
| DeRCFR | $2.88 \pm 0.06$ | $2.61 \pm 0.06$ | $2.87 \pm 0.05$ | $3.07 \pm 0.06$ | $3.11 \pm 0.07$ | $3.04 \pm 0.06$ | $4.77 \pm 0.11$ | $4.50 \pm 0.10$ |
| DESCN | $2.93 \pm 0.11$ | $2.66 \pm 0.09$ | $3.27 \pm 0.81$ | $3.46 \pm 0.79$ | $3.12 \pm 0.20$ | $3.06 \pm 0.20$ | $4.91 \pm 0.37$ | $4.59 \pm 0.35$ |
| ESCFR | $2.87 \pm 0.08$ | $2.62 \pm 0.07$ | $2.94 \pm 0.08$ | $3.15 \pm 0.09$ | $3.03 \pm 0.09$ | $3.06 \pm 0.09$ | $4.71 \pm 0.15$ | $4.43 \pm 0.15$ |
| CFQP | $2.91 \pm 0.09$ | $2.67 \pm 0.11$ | $3.14 \pm 0.30$ | $3.40 \pm 0.37$ | $3.21 \pm 0.12$ | $3.18 \pm 0.11$ | $4.93 \pm 0.14$ | $4.55 \pm 0.13$ |
| Quantile-Reg | $2.80 \pm 0.06$ | $2.54 \pm 0.05$ | $2.78 \pm 0.08$ | $3.05 \pm 0.09$ | $2.92 \pm 0.07$ | $3.01 \pm 0.08$ | $4.39 \pm 0.13$ | $4.12 \pm 0.10$ |
| Ours | $\mathbf{2.41 \pm 0.58}$ | $\mathbf{2.25 \pm 0.48}$ | $\mathbf{2.25 \pm 0.07}$ | $\mathbf{2.33 \pm 0.07}$ | $\mathbf{2.51 \pm 0.07}$ | $\mathbf{2.46 \pm 0.06}$ | $\mathbf{3.78 \pm 0.61}$ | $\mathbf{3.61 \pm 0.56}$ |

Proposition 5.3 indicates that $\hat{R}_{x'}(t|x, z, y)$ is an asymptotically unbiased estimator of $R_{x'}(t|x, z, y)$, demonstrating the validity of the estimator of the ideal loss. The loss $\hat{R}_{x'}(t|x, z, y)$ is applicable only for discrete treatments due to the terms $\mathbb{I}(x_k = x')$ and $\mathbb{I}(x_k = x)$. However, it can be easily extended to continuous treatments, as detailed in Appendix C.

# 6 EXPERIMENTS

## 6.1 SYNTHETIC EXPERIMENT

**Simulation Process.** We generate the synthetic dataset by the following process. First, we sample the covariate $Z \sim \mathcal{N}(0, I_m)$ and the treatment $X \sim \text{Bern}(\pi(Z))$, where $\text{Bern}(\cdot)$ is the Bernoulli distribution with probability $\pi(Z) = \mathbb{P}(X = 1 \mid Z) = \sigma(W_x \cdot Z)$, $\sigma(\cdot)$ is the sigmoid function, and $W_x \sim \text{Unif}(-1, 1)^m$, $\text{Unif}(\cdot)$ is the uniform distribution. Then, we sample the noise $U_0 \sim \mathcal{N}(0, 1)$ and $U_1 = \alpha \cdot U_0$ to consider the heterogeneity of the exogenous variables, where $\alpha$ is the hyper-parameter to control the heterogeneity degree. Finally, we simulate $Y_1 = W_y \cdot Z + U_1$ and $Y_0 = W_y \cdot Z/\alpha + U_0$ with $W_y \sim \mathcal{N}(0, I_m)$. We generate 10,000 samples with 63/27/10 train/validation/test split and vary $m \in \{5, 10, 20, 40\}$ in our synthetic experiment.

**Baselines and Evaluation Metrics.** We compare our method with the following baselines: T-learner (Künzel et al., 2019), X-learner (Künzel et al., 2019), BNN (Johansson et al., 2016), TARNet (Shalit et al., 2017), CFRNet (Shalit et al., 2017), CEVAE (Louizos et al., 2017), DragonNet (Shi et al., 2019), DeRCFR (Wu et al., 2022), DESCN (Zhong et al., 2022), ESCFR (Wang et al., 2023), CFQP (Brouwer, 2022), and Quantile-Reg (Xie et al., 2023). Following the previous studies (Shalit et al., 2017; Yao et al., 2018), we evaluation the individual treatment effect estimation by using the *Precision in Estimation of Heterogeneous Effects* (PEHE) as $\epsilon_{\text{PEHE}} = \frac{1}{N} \sum_{i=1}^{N} ((\hat{Y}_1(Z_i) - \hat{Y}_0(Z_i)) - (Y_i(1) - Y_i(0)))^2$, where $\hat{Y}_1(Z)$ and $\hat{Y}_0(Z)$ are the predicted values for the corresponding true potential outcomes. Both in-sample and out-of-sample performances are reported in our experiments. See Appendix D for more details.

**Performance Analysis.** The results of estimation performance are shown in Table 1. First, the Quantile-Reg method achieves the most competitive performance across all baselines. Second, our method stably outperforms all baselines with varying covariate dimensions $m$, benefiting from the robustness of our estimation method. In addition, we compared our method with the Quantile-Reg under varying heterogeneity degrees and the results are shown in Figure 1. As the heterogeneity degree increases, our method stably outperforms the Quantile-Reg in terms of PEHE. Moreover, as shown in Figure 2, our method stably outperforms the Quantile-Reg and ESCFR methods with different kernels and bandwidths, which further verifies the superiority of our method.

## 6.2 REAL-WORLD EXPERIMENT

**Dataset and Prepossessing.** Following previous studies Shalit et al. (2017),Louizos et al. (2017),Yoon et al. (2018), and Yao et al. (2018), we conduct experiments on one semi-synthetic

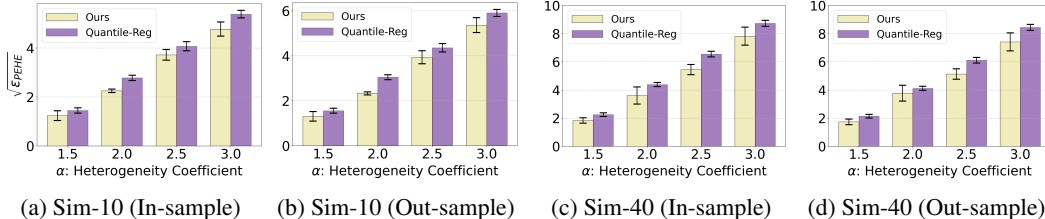

(a) Sim-10 (In-sample)  (b) Sim-10 (Out-sample)  (c) Sim-40 (In-sample)  (d) Sim-40 (Out-sample)

Figure 1: Estimation performance of individual treatment effects under varying heterogeneity degrees.

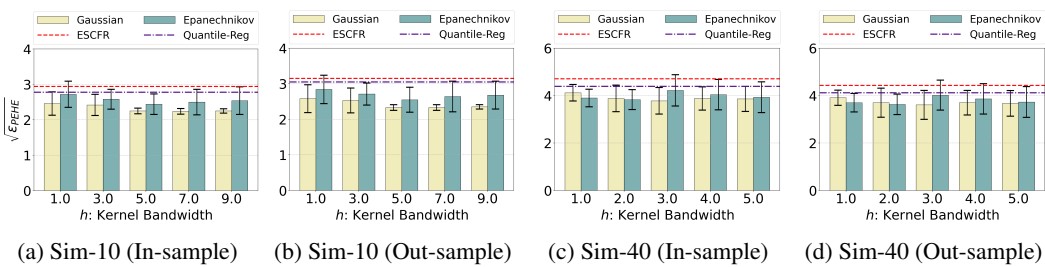

(a) Sim-10 (In-sample)  (b) Sim-10 (Out-sample)  (c) Sim-40 (In-sample)  (d) Sim-40 (Out-sample)

Figure 2: The estimation performance with different kernels and bandwidths.

dataset, IHDP, and one real-world dataset, JOBS. The IHDP dataset (Hill, 2011) is constructed from the Infant Health and Development Program (IHDP) with 747 individuals and 25 covariates. The JOBS dataset (LaLonde, 1986) is based on the National Supported Work program with 3,212 individuals and 17 covariates. We follow Shalit et al. (2017) to split the data into training/validation/testing set with ratios 63/27/10 and 56/24/20 for the IHDP and the JOBS dataset, respectively. We repeat the experiment 100 times for the IHDP dataset and 10 times for the JOBS dataset.

**Evaluation Metrics.** Following previous studies Shalit et al. (2017); Louizos et al. (2017); Yao et al. (2018), the absolute error in *Average Treatment Effect* (ATE) is defined as $\epsilon_{\text{ATE}} = \frac{1}{N}|\sum_{i=1}^{N}((\hat{Y}_1(Z_i) - \hat{Y}_0(Z_i)) - (Y_i(1) - Y_i(0)))|$, and we use $\sqrt{\epsilon_{\text{PEHE}}}$ and $\epsilon_{\text{ATE}}$ to evaluate performance on the IHDP dataset. For the JOBS dataset, since one of the potential outcomes is not available, we evaluate the performance using the absolute error in *Average Treatment effect on the Treated* (ATT) as $\epsilon_{\text{ATT}} = |\text{ATT} - \frac{1}{|T|}\sum_{i \in T}(\hat{Y}_1(Z_i) - \hat{Y}_0(Z_i)|$ with $\text{ATT} = |\frac{1}{|T|}\sum_{i \in T} Y_i - \frac{1}{|C \cap E|}\sum_{i \in C \cap E} Y_i|$. We also use the policy risk $R_{\text{Pol}} = 1 - (\mathbb{E}[Y(1) \mid \hat{Y}_1(Z) - \hat{Y}_0(Z) > 0, X = 1] \cdot \mathbb{P}(\hat{Y}_1(Z) - \hat{Y}_0(Z) > 0) + \mathbb{E}[Y(0) \mid \hat{Y}_1(Z) - \hat{Y}_0(Z) \leq 0, X = 0] \cdot \mathbb{P}(\hat{Y}_1(Z) - \hat{Y}_0(Z) \leq 0))$, where $T, C, E$ are the treatment sample set, control sample set, and randomized sample set, respectively.

**Performance Comparison.** The experiment results are shown in Table 2. Similar to the synthetic experiment, the Quantile-Reg method still achieves the most competitive performance compared to the other baselines. Our method stably outperforms all the baselines on both the semi-synthetic dataset IHDP and the real-world dataset JOBS, especially in the out-sample scenario. This provides the empirical evidence of the effectiveness of our method.

## 7 RELATED WORK

**Conditional Average Treatment Effect (CATE).** CATE also referred to as heterogeneous treatment effect, represents the average treatment effects on subgroups categorized by covariate values, and plays a central role in areas such as precision medicine Kosorok & Laber (2019) and policy learning Dudík et al. (2011). Benefiting from recent advances in machine learning, many methods have been proposed for estimating CATE, including matching methods Rosenbaum & Rubin (1983); Schwab et al. (2018); Yao et al. (2018), tree-based methods Chipman et al. (2010); Wager & Athey (2018), representation learning methods Johansson et al. (2016); Shalit et al. (2017); Shi et al. (2019); Wu et al. (2022); Wang et al. (2023), and generative methods Louizos et al. (2017); Yoon et al. (2018). Unlike the existing work devoted to estimating CATE at the intervention level for

Table 2: The experiment results on the IHDP dataset and JOBS dataset. The best result is bolded.

| | IHDP | | | | JOBS | | | |
| --- | --- | --- | --- | --- | --- | --- | --- | --- |
| | In-sample | | Out-sample | | In-sample | | Out-sample | |
| Methods | $\sqrt{\epsilon_{PEHE}}$ | $\epsilon_{ATE}$ | $\sqrt{\epsilon_{PEHE}}$ | $\epsilon_{ATE}$ | $R_{Pol}$ | $\epsilon_{ATT}$ | $R_{Pol}$ | $\epsilon_{ATT}$ |
| T-learner | $1.49 \pm 0.03$ | $0.37 \pm 0.05$ | $1.81 \pm 0.04$ | $0.49 \pm 0.04$ | $0.31 \pm 0.06$ | $0.16 \pm 0.10$ | $0.27 \pm 0.08$ | $0.20 \pm 0.07$ |
| X-learner | $1.50 \pm 0.02$ | $0.21 \pm 0.05$ | $1.73 \pm 0.03$ | $0.36 \pm 0.07$ | $0.16 \pm 0.04$ | $0.07 \pm 0.05$ | $0.16 \pm 0.03$ | $0.10 \pm 0.09$ |
| BNN | $2.09 \pm 0.16$ | $1.00 \pm 0.23$ | $2.37 \pm 0.15$ | $1.18 \pm 0.19$ | $0.15 \pm 0.01$ | $0.08 \pm 0.03$ | $0.16 \pm 0.02$ | $0.13 \pm 0.07$ |
| TARNet | $1.52 \pm 0.07$ | $0.22 \pm 0.13$ | $1.78 \pm 0.07$ | $0.34 \pm 0.18$ | $0.17 \pm 0.06$ | $0.06 \pm 0.08$ | $0.18 \pm 0.09$ | $0.10 \pm 0.06$ |
| CFRNet | $1.46 \pm 0.06$ | $0.17 \pm 0.15$ | $1.77 \pm 0.06$ | $0.32 \pm 0.20$ | $0.17 \pm 0.03$ | $\mathbf{0.05 \pm 0.03}$ | $0.19 \pm 0.07$ | $0.10 \pm 0.04$ |
| CEVAE | $4.08 \pm 0.88$ | $3.67 \pm 1.23$ | $4.12 \pm 0.91$ | $3.75 \pm 1.23$ | $0.18 \pm 0.05$ | $0.09 \pm 0.03$ | $0.22 \pm 0.08$ | $0.10 \pm 0.09$ |
| DragonNet | $1.49 \pm 0.08$ | $0.22 \pm 0.14$ | $1.80 \pm 0.06$ | $0.29 \pm 0.19$ | $0.17 \pm 0.06$ | $0.07 \pm 0.07$ | $0.20 \pm 0.08$ | $0.11 \pm 0.09$ |
| DeRCFR | $1.48 \pm 0.04$ | $0.25 \pm 0.14$ | $1.69 \pm 0.06$ | $0.25 \pm 0.14$ | $0.15 \pm 0.02$ | $0.14 \pm 0.04$ | $0.16 \pm 0.04$ | $0.15 \pm 0.11$ |
| DESCN | $2.08 \pm 0.98$ | $0.74 \pm 1.00$ | $2.67 \pm 1.45$ | $1.04 \pm 1.46$ | $0.15 \pm 0.02$ | $0.21 \pm 0.14$ | $0.22 \pm 0.16$ | $0.16 \pm 0.04$ |
| ESCFR | $1.46 \pm 0.09$ | $0.16 \pm 0.16$ | $1.73 \pm 0.08$ | $0.27 \pm 0.16$ | $0.14 \pm 0.02$ | $0.10 \pm 0.03$ | $0.15 \pm 0.02$ | $0.10 \pm 0.08$ |
| Quantile-Reg | $1.43 \pm 0.05$ | $0.14 \pm 0.09$ | $1.56 \pm 0.03$ | $0.18 \pm 0.09$ | $0.14 \pm 0.01$ | $0.06 \pm 0.01$ | $0.15 \pm 0.01$ | $0.07 \pm 0.04$ |
| CFQP | $1.47 \pm 0.10$ | $0.18 \pm 0.17$ | $1.48 \pm 0.05$ | $0.15 \pm 0.08$ | $0.15 \pm 0.02$ | $0.23 \pm 0.15$ | $0.16 \pm 0.03$ | $0.15 \pm 0.07$ |
| Ours | $\mathbf{1.41 \pm 0.02}$ | $\mathbf{0.11 \pm 0.10}$ | $\mathbf{1.50 \pm 0.06}$ | $\mathbf{0.13 \pm 0.08}$ | $\mathbf{0.08 \pm 0.04}$ | $0.06 \pm 0.02$ | $\mathbf{0.11 \pm 0.05}$ | $\mathbf{0.05 \pm 0.05}$ |

subgroups, our work focuses on counterfactual inference at the more challenging and fine-grained individual level.

**Counterfactual Inference.** Counterfactual inference involves the identification and estimation of counterfactual outcomes. For identification, Shpitser & Pearl (2007) provided an algorithm leveraging counterfactual graphs to identify counterfactual queries. In addition, Correa et al. (2021) discussed the identifiability of nested counterfactuals within a given causal graph. More relevant to our work, Lu et al. (2020) and Xie et al. (2023) studied the identifiability assumptions in the setting of backdoor criterion under homogeneity and strict monotonicity assumptions. Several methods focus on determining its bounds with less stringent assumptions, such as Balke & Pearl (1994), Tian & Pearl (2000), Pearl (2009), Pearl et al. (2016), Finkelstein & Shpitser (2020), Zhang et al. (2022), and Melnychuk et al. (2023).

For estimation, Pearl et al. (2016) introduced a three-step procedure for counterfactual inference. Many machine learning methods estimate counterfactual outcomes in this framework, such as Lu et al. (2020), Mesnard et al. (2021), Brouwer (2022), Shah et al. (2022), Yan et al. (2023a), Nasr-Esfahany et al. (2023) and Chao et al. (2023). Recently, Xie et al. (2023) employed quantile regression to estimate the counterfactual outcomes, effectively circumventing the need for SCM estimation. In our work, we extend the above methods in both identification and estimation.

Recently, counterfactual inference methods have been extensively applied across various application scenarios, such as counterfactual fairness (Kusner et al., 2017; Zuo et al., 2023; Anthis & Veitch, 2023; Kavouras et al., 2023; Chen et al., 2023a), policy evaluation and improvement (Tang & Wiens, 2023; Saveski et al., 2023; Chen et al., 2023b), reinforcement learning (Lu et al., 2020; Tsirtsis & Rodriguez, 2023; Liu et al., 2023a; Shao et al., 2023; Meulemans et al., 2023; Haugh & Singal, 2023; Zenati et al., 2023), imitaion learning (Sun et al., 2023), counterfactual generation (Yan et al., 2023b; Prabhu et al., 2023; Feder et al., 2023; Ribeiro et al., 2023), counterfactual explanation (Kenny & Huang, 2023; Raman et al., 2023; Hamman et al., 2023; Wu et al., 2023; Ley et al., 2023), counterfactual harm (Richens et al., 2022; Li et al., 2023), physical audiovisual commonsense reasoning (Lv et al., 2023), interpretable time series prediction (Yan & Wang, 2023), classification and detection in medical imaging (Fontanella et al., 2023), data valuation (Liu et al., 2023b), etc. Therefore, developing novel counterfactual inference methods holds significant practical implications.

## 8 CONCLUSION

This work addresses the fundamental challenge of counterfactual inference in the absence of a known SCM and under heterogeneous endogenous variables. We first introduce the rank preservation assumption to identify counterfactual outcomes. Then, we propose a novel ideal loss for unbiased learning of counterfactual outcomes and develop a kernel-based estimator for practical implementation. The convexity of the ideal loss and the unbiased nature of the proposed estimator contribute to the robustness and reliability of our method. A potential limitation arises when the propensity score is extremely small in certain data sparsity scenarios, which may cause instability in the estimation method. Further investigation is warranted to address and overcome this challenge.

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

# A    PROOFS IN SECTIONS 3 AND 4

One can show Lemma 3.3 by a similar argument of the proof of Theorem 1 in Xie et al. (2023). For the sake of self-containedness, we provide a novel proof of it.

**Lemma 3.3** *Under Assumptions 3.1-3.2, $y_{x'}$ is identifiable.*

*Proof of Lemma 3.3.* First, the distributions $\mathbb{P}(Y_x|Z = z)$ and $\mathbb{P}(Y_{x'}|Z = z)$ can be identified as $\mathbb{P}(Y|X = x, Z = z)$ and $\mathbb{P}(Y|X = x', Z = z)$, respectively, by the backdoor criterion (i.e., $(Y_x, Y_{x'}) \perp\!\!\!\perp X|Z$) of the setting.

Then, according to the model (1), we can equivalently write
$$Y_x = f_Y(x, z, U_x), \ Y_{x'} = f_Y(x', z, U_{x'}),$$
and $Y$ and $U_X$ in model (1) can be expressed as $Y = \sum_{x \in \mathcal{X}} \mathbb{I}(X = x) \cdot Y_x$ and $U_X = \sum_{x \in \mathcal{X}} \mathbb{I}(X = x) \cdot U_x$, where $\mathcal{X}$ is the support set of $X$ and $\mathbb{I}(\cdot)$ is an indicator function. Assumption 3.1 implies that $U_X = U_x = U_{x'}$ conditional on $Z$, i.e., $Y_x = f_Y(x, z, U_X), \ Y_{x'} = f_Y(x', z, U_X)$.

Finally, for the individual with observation $(X = x, Z = z, Y = y)$, we denote $(y_x, y_{x'})$ as the true values of $(Y_x, Y_{x'})$ for this individual. For this individual, we can identify the quantile of $y_x$ in the distribution of $\mathbb{P}(Y_x|Z = z) = \mathbb{P}(Y|X = x, Z = z)$, denoted by $\tau^*$. Let $u_{\tau^*}$ be the true value of $U_X$ for this individual, it is the $\tau^*$-quantile in the distribution $\mathbb{P}(U_X|Z = z)$, then we have
$$
\begin{aligned}
\tau^* &= \mathbb{P}(Y_x \leq y_x|Z = z) && \text{(by the definition of } \tau) \\
&= \mathbb{P}(U_x \leq u_\tau|Z = z) && \text{(by Assumption 3.2)} \\
&= \mathbb{P}(U_{x'} \leq u_\tau|Z = z) && \text{(by Assumption 3.1)} \\
&= \mathbb{P}(Y_{x'} \leq f_Y(x', z, u_{\tau^*})|Z = z) && \text{(by Assumption 3.2)} \\
&= \mathbb{P}(Y_{x'} \leq y_{x'}|Z = z) && \text{(by the definition of } y_{x'}),
\end{aligned}
$$
which implies that for this individual, its rankings of $y_x$ and $y_{x'}$ are the same in the distributions of $\mathbb{P}(Y_x|Z = z)$ and $\mathbb{P}(Y_{x'}|Z = z)$, resepcctively. Thus, $y_{x'}$ is identified as the $\tau^*$-quantile of the distribution $\mathbb{P}(Y_{x'}|Z = z) = \mathbb{P}(Y|X = x', Z = z)$.

$\square$

**Proposition 4.3** *Under Assumptions 4.2, $y_{x'}$ is identifiable.*

*Proof of Proposition 4.3.* For the individual with observation $(X = x, Z = z, Y = y)$, we denote $(y_x, y_{x'})$ as the true values of $(Y_x, Y_{x'})$. Assumption 4.2 implies that for this individual, its rankings of $y_x$ and $y_{x'}$ are the same in the distributions of $\mathbb{P}(Y_x|Z = z)$ and $\mathbb{P}(Y_{x'}|Z = z)$, respectively. Therefore,
$$\mathbb{P}(Y_x \leq y_x|Z = z) = \mathbb{P}(Y_{x'} \leq y_{x'}|Z = z). \tag{3}$$

Since $y_x = y$ is observed and the distributions $\mathbb{P}(Y_x|Z = z)$ and $\mathbb{P}(Y_{x'}|Z = z)$ can be identified as $\mathbb{P}(Y|X = x, Z = z)$ and $\mathbb{P}(Y|X = x', Z = z)$, respectively, by the backdoor criterion (i.e., $(Y_x, Y_{x'}) \perp\!\!\!\perp X|Z$), we can identify the quantile of $y_x$ in the distribution of $\mathbb{P}(Y|X = x, Z = z)$, denoted by $\tau^*$. Then
$$\mathbb{P}(Y_{x'} \leq y_{x'}|Z = z) = \tau^*,$$
which yields that $\theta$ is identified as the $\tau^*$-quantile of $\mathbb{P}(Y|X = x', Z = z)$.

$\square$

**Proposition 4.4** Under Assumption 3.1, or more generally, if $U_x$ is a strictly monotone increasing function of $U_{x'}$, Assumption 4.2 is equivalent to Assumption 3.2.

*Proof of Proposition 4.4.* According to the model (1), we can equivalently write
$$Y_x = f_Y(x, z, U_x), \ Y_{x'} = f_Y(x', z, U_{x'}).$$

Suppose that $U_x$ is a strictly monotone increasing function of $U_{x'}$ (Assumption 3.1, i.e., $U_x = U_{x'}$, is a special case of it). Under this condition, we next prove sufficiency and necessity, respectively.

First, we show that Assumption 3.2 implies Assumption 4.2. If Assumption 3.2 holds, then $Y_x$ is a strictly monotonic function of $U_x$, and $Y_{x'}$ is a strictly monotonic function of $U_{x'}$. Since $U_x$ is a strictly monotone increasing function of $U_{x'}$, then $Y_x$ is a strictly increasing monotonic function of $Y_{x'}$, which leads to Assumption 4.2.

Second, we show that Assumption 4.2 implies Assumption 3.2. If Assumption 4.2 holds, then given $Z = z$, $Y_x$ is a strictly increasing function of $Y_{x'}$. When $U_x$ is a strictly monotone increasing function of $U_{x'}$ and note that

$$Y_x = f_Y(x, z, U_X), \ Y_{x'} = f_Y(x', z, U_X),$$

which implies that $f_Y$ is a strictly monotonic function of $U_X$, i.e., Assumption 3.2 holds.

This finishes the proof. $\qquad\square$

**Proposition 4.7** *Under Assumption 4.6, the conclusion in Proposition 4.3 also holds.*

*Proof of Proposition 4.7.* This can be shown through a proof analogous to that of Proposition 4.3.

$\qquad\square$

# B    PROOFS IN SECTION 5

Recall that $l_\tau(\xi) = \tau\xi \cdot \mathbb{I}(\xi \geq 0) + (\tau - 1)\xi \cdot \mathbb{I}(\xi < 0)$, and

$$q(x, z; \tau) \triangleq \inf_y \{y : \mathbb{P}(Y \leq y | X = x, Z = z) \geq \tau\}$$

$$q_0(z; \tau) \triangleq \inf_y \{y : \mathbb{P}(Y_0 \leq y | Z = z) \geq \tau\}$$

$$q_1(z; \tau) \triangleq \inf_y \{y : \mathbb{P}(Y_1 \leq y | Z = z) \geq \tau\}.$$

**Proposition 5.1** We have that

*(i) $q_x(Z; \tau) = \arg\min_f \mathbb{E}[l_\tau(Y_x - f(Z))]$ for $x = 0, 1$;*

*(ii) $q(X, Z; \tau) = \arg\min_f \mathbb{E}[l_\tau(Y - f(X, Z))]$.*

*Proof of Proposition 5.1.* We prove $q_x(Z; \tau) = \arg\min_f \mathbb{E}[l_\tau(Y_x - f(Z))]$, and $q(X, Z; \tau) = \arg\min_f \mathbb{E}[l_\tau(Y - f(X, Z))]$ can be derived by an exactly similar manner. We write

$$\mathbb{E}[l_\tau(Y_x - f(Z))] = \mathbb{E}[\mathbb{E}\{l_\tau(Y_x - f(Z)) \mid Z\}].$$

To obtain the conclusion, note that $l_\tau(Y_x - f(Z))$ is always positive, it suffices to show that

$$q_x(z; \tau) = \arg\min_f \mathbb{E}[l_\tau(Y_x - f(Z)) \mid Z = z] \tag{4}$$

for any given $Z = z$. Next, we focus on analyzing the term $\mathbb{E}[l_\tau(Y_x - f(Z)) \mid Z = z]$. Given $Z = z$, $f(Z)$ is a constant and we denote it by $c$, then

$$\mathbb{E}[l_\tau(Y_x - f(Z)) \mid Z = z]$$
$$= \mathbb{E}[l_\tau(Y_x - c) \mid Z = z]$$
$$= \mathbb{E}\Big[\tau(Y_x - c)\mathbb{I}(Y_x \geq c) + (\tau - 1)(Y_x - c)\mathbb{I}(Y_x < c) \mid Z = z\Big]$$
$$= \tau \int_c^\infty (y_x - c)g(y_x|z)dy_x + (\tau - 1)\int_{-\infty}^c (y_x - c)g(y_x|z)dy_x,$$

where $g(y_x|z)$ denotes the probability density function of $Y_x$ given $Z = z$.

Since the check function is a convex function, differentiating $\mathbb{E}[l_\tau(Y_x - c) \mid Z = z]$ with respect to $c$ and setting the derivative to zero will yield the solution for the minimum

$$\frac{\partial}{\partial c}\mathbb{E}[l_\tau(Y_x - c) \mid Z = z]$$

$$= \tau \int_c^\infty \frac{\partial}{\partial c}[(y_x - c)g(y_x|z)]dy_x + (\tau - 1)\int_{-\infty}^c \frac{\partial}{\partial c}[(y_x - c)g(y_x|z)]dy_x$$

$$= -\tau\left(1 - \int_{-\infty}^c g(y_x|z)dy_x\right) + (1 - \tau)\int_{-\infty}^c g(y_x|z)dy_x.$$

Then let $\frac{\partial}{\partial c}\mathbb{E}[l_\tau(Y_x - c) \mid Z = z] = 0$ leads to that

$$\int_{-\infty}^c g(y_x|z)dy_x = \tau,$$

that is, $c = q_x(z; \tau)$. This completes the proof of Proposition 5.1.

$\square$

**Theorem 5.2.** *If the probability density function of $Y$ given $Z$ is continuous, then the loss $R_{x'}(t; x, z, y)$ is minimized uniquely at $t^*$ satisfying*

$$\mathbb{P}(Y_{x'} \leq t^*|Z = z) = \mathbb{P}(Y_x \leq y|Z = z).$$

*Proof of Theorem 5.2.* Recall that

$$R_{x'}(t|x, z, y) = \mathbb{E}\left[|Y_{x'} - t| \,\Big|\, Z = z\right] + \mathbb{E}\left[\text{sign}(Y_x - y) \,\Big|\, Z = z\right] \cdot t.$$

Let $g(y_x|z)$ be the probability density function of $Y_x$ given $Z = z$. By calculation,

$$\mathbb{E}\left[|Y_{x'} - t| \,\Big|\, Z = z\right] = \int_t^\infty (y_{x'} - t)g(y_{x'}|z)dy_{x'} + \int_{-\infty}^t (t - y_{x'})g(y_{x'}|z)dy_{x'},$$

$$\frac{\partial}{\partial t}\mathbb{E}\left[|Y_{x'} - t| \,\Big|\, Z = z\right] = -\left(1 - \int_{-\infty}^t g(y_{x'}|z)dy_{x'}\right) + \int_{-\infty}^t g(y_{x'}|z)dy_{x'} = 2\mathbb{P}(Y_{x'} \leq t|Z = z) - 1,$$

and

$$\mathbb{E}\left[\text{sign}(Y_x - y) \,\Big|\, Z = z\right] = \mathbb{E}\left[-2\mathbb{I}(Y_x \leq y) + 1 \,\Big|\, Z = z\right] = -2\mathbb{P}(Y_x \leq y|Z = z) + 1,$$

we have

$$\frac{\partial}{\partial t}R_{x'}(t|x, z, y) = 2\mathbb{P}(Y_{x'} \leq t|Z = z) - 1 + \mathbb{E}\left[\text{sign}(Y - y) \,\Big|\, Z = z\right]$$

$$= 2\mathbb{P}(Y_{x'} \leq t|Z = z) - 1 - 2\mathbb{P}(Y_x \leq y|Z = z) + 1$$

$$= 2\left\{\mathbb{P}(Y_{x'} \leq t|z) - \mathbb{P}(Y_x \leq y|z)\right\}.$$

Since

$$\frac{\partial^2}{\partial t^2}R_{x'}(t|x, z, y) = 2\partial\mathbb{P}(Y_{x'} \leq t|z)/\partial t = 2g(y_{x'} = t|z) \geq 0,$$

$R_{x'}(t|x, z, y)$ is a convex function with respect to $t$. Letting $\frac{\partial}{\partial t}R_{x'}(t|x, z, y) = 0$ yields that

$$\mathbb{P}(Y_{x'} \leq t|z) - \mathbb{P}(Y_x \leq y|z) = 0.$$

That is, $R_{x'}(t|x, z, y)$ attains its minimum at $t = q_{x'}(z; \tau^*)$, where $\tau^*$ is the quantile of $y$ in the distribution $\mathbb{P}(Y_x|Z = z)$.

$\square$

**Proposition 5.3.** *If $h \to 0$ as $N \to \infty$, and the density function of $Z$ twice differentiable, then*

$$\hat{R}_{x'}(t; x, z, y) \xrightarrow{\mathbb{P}} R_{x'}(t; x, z, y),$$

*where $\xrightarrow{\mathbb{P}}$ means convergence in probability.*

*Proof of Proposition 5.3.* For analyzing the theoretical properties of $\hat{R}_{x'}(t; x, z, y)$, we rewritten $\hat{R}_{x'}(t; x, z, y)$ as

$$\hat{R}_{x'}(t; x, z, y) = \frac{\sum_{k=1}^{N} K_h(Z_k - z)\frac{\mathbb{I}(X_k = x')}{p_{x'}(Z_k)}|Y_k - t|}{\sum_{k=1}^{N} K_h(Z_k - z)} + \frac{\sum_{k=1}^{N} K_h(Z_k - z)\frac{\mathbb{I}(X_k = x)}{p_x(Z_k)} \cdot \text{sign}(Y_k - y)}{\sum_{i=1}^{N} K_h(Z_k - z)} \cdot t,$$

where the capital letters denote random variables and lowercase letters denote their realizations. This is slightly different from that used in the main text.

To show the conclusion, it is sufficient to prove that

$$\frac{\sum_{k=1}^{N} K_h(Z_k - z)\frac{\mathbb{I}(X_k = x')}{p_{x'}(Z_k)}|Y_k - t|}{\sum_{k=1}^{N} K_h(Z_k - z)} \xrightarrow{\mathbb{P}} \mathbb{E}\left[\frac{\mathbb{I}(X = x')}{p_{x'}(z)}|Y - t| \,\Big|\, Z = z\right] = \mathbb{E}\left[|Y_{x'} - t| \,\Big|\, Z = z\right],$$
(5)

$$\frac{\sum_{k=1}^{N} K_h(Z_k - z)\frac{\mathbb{I}(X_k = x)}{p_x(Z_k)} \cdot \text{sign}(Y_k - y)}{\sum_{i=1}^{N} K_h(Z_k - z)} \xrightarrow{\mathbb{P}} \mathbb{E}\left[\frac{\mathbb{I}(X = x)}{p_x(z)} \cdot \text{sign}(Y - y) \,\Big|\, Z = z\right] = \mathbb{E}\left[\text{sign}(Y_x - y) \,\Big|\, Z = z\right].$$
(6)

We prove equation (5) only, as equation (6) can be addressed similarly.

Note that

$$\frac{\sum_{k=1}^{N} K_h(Z_k - z)\frac{\mathbb{I}(X_k = x')}{p_{x'}(Z_k)}|Y_k - t|}{\sum_{k=1}^{N} K_h(Z_k - z)} = \frac{\frac{1}{N}\sum_{k=1}^{N} K_h(Z_k - z)\frac{\mathbb{I}(X_k = x')}{p_{x'}(Z_k)}|Y_k - t|}{\frac{1}{N}\sum_{k=1}^{N} K_h(Z_k - z)},$$

we analyze the denominator and numerator on the right side of the equation separately. For the denominator, it is an average of $N$ independent random variables and converges to its expectation $\mathbb{E}[K_h(Z_k - z)]$ almost surely. Let $g(z_k)$ be the probability density function of $Z_k$, and $g^{(1)}(z_k)$ is its first derivative. Since

$$\mathbb{E}[K_h(Z_k - z)] = \int \frac{1}{h} K(\frac{z_k - z}{h}) g(z_k) dz_k$$

$$= \int K(u) g(z + hu) du \qquad \text{(let } z_k = z + hu)$$

$$= \int K(u) \cdot \{g(z) + g^{(1)}(z)hu + o(h)\} du \qquad \text{(by Taylor Expansion)}$$

$$= g(z) \int K(u) du + g^{(1)}(z)h \int K(u)u du + o(h)$$

$$= g(z) + o(h) \qquad \text{(by the definition of kernel function)}, \tag{7}$$

when $h \to 0$ as $N \to \infty$, the denominator converges to $g(z)$ in probability.

Next, we focus on dealing with the numerator, which also converges to its expectation.

$$\mathbb{E}[K_h(Z_k - z)\frac{\mathbb{I}(X_k = x')}{p_{x'}(Z_k)}|Y_k - t|]$$

$$= \mathbb{E}\left[K_h(Z_k - z)\mathbb{E}\left\{\frac{\mathbb{I}(X_k = x')}{p_{x'}(Z_k)}|Y_k - t|\Big|Z_k\right\}\right] \qquad \text{(by the law of iterated expectations)}$$

$$= \mathbb{E}\left[K_h(Z_k - z)\mathbb{E}\left\{\frac{\mathbb{I}(X_k = x')}{p_{x'}(Z_k)}|Y_{x',k} - t|\Big|Z_k\right\}\right] \qquad \text{(write } Y_k \text{ as the form of potential outcome)}$$

$$= \mathbb{E}\left[K_h(Z_k - z)\mathbb{E}\left\{|Y_{x',k} - t|\Big|Z_k\right\}\right] \qquad \text{(by backdoor criterion } Y_{x',k} \perp\!\!\!\perp X_k|Z_k). \tag{8}$$

Define $m(Z) = \mathbb{E}[|Y_{x'} - t| | Z]$ and $m^{(1)}(Z)$ is its first derivative, then the right side of equation (5) is $m(z)$, and

$$\mathbb{E}\Big[K_h(Z_k - z) \cdot \mathbb{E}\big\{|Y_{x',k} - t| \big| Z_k\big\}\Big] = \mathbb{E}\Big[K_h(Z_k - z) \cdot m(Z_k)\Big]$$

$$= \int \frac{1}{h} K(\frac{z_k - z}{h}) \cdot m(z_k) \cdot g(z_k) dz_k$$

$$= \int K(u) \cdot m(z + hu) \cdot g(z + hu) du \qquad \text{(let } z_k = z + hu)$$

$$= \int K(u) \cdot \{m(z) + m^{(1)}(z)hu + o(h)\} \cdot \{g(z) + g^{(1)}(z)hu + o(h)\} du \qquad \text{(by Taylor Expansion)}$$

$$= m(z)g(z) + o(h). \tag{9}$$

Thus, when $h \to 0$ as $N \to \infty$, the numerator converges to $g(z)$ in probability.

Combining equations (7), (8), and (9) yields the equality (5). This completes the proof.

$\square$

## C    EXTENSION TO CONTINUOUS OUTCOME

When the treatment is continuous, we can estimate the ideal loss with the following estimator

$$\tilde{R}_{x'}(t|x, z, y) = \frac{\sum_{k=1}^N K_h(z_k - z)\frac{K_h(x_k - x')}{p_{x'}(z_k)}|y_k - t|}{\sum_{k=1}^N K_h(z_k - z)} + \frac{\sum_{k=1}^N K_h(z_k - z)\frac{K_h(x_k - x)}{p_x(z_k)} \cdot \text{sign}(y_k - y)}{\sum_{k=1}^N K_h(z_k - z)} \cdot t,$$

which is a smoothed version of the estimator

$$\hat{R}_{x'}(t|x, z, y) = \frac{\sum_{k=1}^N K_h(z_k - z)\frac{\mathbb{I}(x_k = x')}{p_{x'}(z_k)}|y_k - t|}{\sum_{k=1}^N K_h(z_k - z)} + \frac{\sum_{k=1}^N K_h(z_k - z)\frac{\mathbb{I}(x_k = x)}{p_x(z_k)} \cdot \text{sign}(y_k - y)}{\sum_{k=1}^N K_h(z_k - z)} \cdot t,$$

defined in Section 5. In addition, by a proof similar to that of Proposition 5.3, we also can show that $\tilde{R}_{x'}(t; x, z, y) \xrightarrow{\mathbb{P}} R_{x'}(t; x, z, y)$.

## D    EXPERIMENT DETAILS

We run all experiments on the Google Colab platform. For the representation model, we use the MLP for the base model and tune the layers in $\{1, 2, 3\}$. In addition, we adopt the logistic regression model as the propensity model. We tune the learning rate in $\{0.001, 0.005, 0.01, 0.05, 0.1\}$. For the kernel choice, we select the kernel function between the Gaussian kernel function and the Epanechnikov kernel function, and tune the bandwidth in $\{1, 3, 5, 7, 9\}$.

