# OpenReview forum: "Counterfactual Learning under Rank Preservation"
_ICLR.cc/2025/Conference — Submitted to ICLR 2025_

### Official Review · Reviewer_md3U · 2024-11-03

**Soundness:** 3
**Presentation:** 3
**Contribution:** 2
**Rating:** 5
**Confidence:** 3

**Summary:**

This paper proposes a new identification condition, under which the individual counterfactual can be identified. The condition requires the rank of counterfactuals to be in accordance with the observed outcomes. The authors then provide an algorithm for learning the counterfactuals via empirical risk minimization. The proposed algorithm is evaluated on synthetic and real data.

**Strengths:**

This paper considers a very challenging and interesting question and offers a thought-provoking solution (the new identification scheme). There is an exhaustive literature review and comparison with existing work.

**Weaknesses:**

If I understand correctly, the new identification condition requires the rank of the observables to be *almost surely* the same as that of the counterfactuals.This appears to be a very strong assumption to me: if patient A has a better baseline than a similar patient B, then the potential outcome of patient A under treatment *has*to be better than that of patient B; but in practice, one could imagine that there can be (at least) some randomness such that this condition is violated. It would be helpful to provide more discussion on why this is a reasonable condition.

**Questions:**

Apart from the comment/question in the weakness section, I have the
following question regarding the learning algorithm guarantee:
1. It appears that Theorem 5.3 applies to a fixed value of t, which does not imply the
guarantee on minimizer of $\hat R_{x'}(t \mid x,z,y)$. If that is the case, I wonder if
the results can be generalized to data-driven t?
2. Does one need to solve an optimization problem for each $(x,z,y)$?

---

### Official Review · Reviewer_UKAh · 2024-11-03

**Soundness:** 4
**Presentation:** 3
**Contribution:** 3
**Rating:** 6
**Confidence:** 4

**Summary:**

Authors study counterfactual inference problem and propose a novel identification/estimation strategy, demonstrating its advantages through existing methods.

**Strengths:**

The paper is well written and organized. Authors clearly list their assumptions and describe their contributions.

The related work is well-covered and it motivates the approach presented in the paper.

The technical contribution is novel. Authors identify an alternative assumption sufficient for identifying the CATE and show how it can be estimated in an unbiased way through a convex loss function.

Experimental evaluation includes many baselines that are prevalent in the literature, and the proposed approach seems to present a notable performance boost.

**Weaknesses:**

The experimental results can benefit from additional discussions, such as why your method has in general higher uncertainty around its performance compared to Quantile Reg (see Table 1 and Figure 1).

Also, it is a bit strange that the results for your method has notably higher standard error for $m=5$. Why is that? Because this would have important implications on methods' utility in worst case, it is important to understand and discuss the reasons behind it.

For instance, are there special data-generating cases where your method performs worse in particular? That does not necessarily put your method down, but it is important to identify and discuss those cases.


Misc--

Line 377 - Typo: "Prepossessing"

Table 1 - CFQP has better out-sample PEHE perf.

**Questions:**

See above.

---

> ### Comment · Reviewer_UKAh · 2024-11-27
> **Concerns not addressed**
>
> As some of my concerns regarding the empirical results were not addressed through authors' rebuttal, I updated my score from 8 to 6, also considering some of the other reviewers' continuing concerns.

---

### Official Review · Reviewer_DfTQ · 2024-11-05

**Soundness:** 2
**Presentation:** 2
**Contribution:** 2
**Rating:** 3
**Confidence:** 3

**Summary:**

The paper introduces a rank preservation assumption to identify counterfactual outcomes. Building on this foundation, it proposes a loss function that yields an unbiased estimator, and further develops a kernel-based estimator for empirical estimation.

**Strengths:**

(1) The paper is clearly written and easy to read. It provides a comprehensive review of previous work on counterfactual learning, identifies gaps, and discusses similarities and differences in assumptions and estimation strategies compared to prior research.

(2)  The proposed loss function does not require prior estimation of the structural causal model, nor does it assume that the two conditional quantile models are identical. Additionally, it does not require explicit estimation of a different quantile value for each individual.

**Weaknesses:**

(1) The main concern is the validity of the rank preservation assumption. Both the identification and estimation methods rely on this key assumption, yet the paper does not provide any testing method or sensitivity analysis for it.

(2) Continuing from (1), we would like to see (i) additional synthetic experiments where rank preservation is violated to assess the performance of the proposed estimation strategy, and (ii) a method for evaluating the rank preservation assumption in real data applications.

(3) The paper only demonstrates consistency for the proposed loss function. Given that the ultimate target estimand is the counterfactual outcome, we would like to see statistical inference results, including consistency and asymptotic variance, for the counterfactual outcome.

(4) The proposed loss function estimation involves kernel smoothing, which may introduce high variance in estimation when covariates are high-dimensional. In the current synthetic experiments, all covariates are independent of each other. We would like to see additional synthetic experiments with high-dimensional, non-independent covariates.

**Questions:**

Please see Weaknesses.

---

### Official Review · Reviewer_vT8b · 2024-11-05

**Soundness:** 2
**Presentation:** 3
**Contribution:** 2
**Rating:** 3
**Confidence:** 3

**Summary:**

The paper proposes a new method for counterfactual estimation. The method needs slightly less stringent assumptions than existing approaches.  Based on a novel convex loss function, the paper proposes a kernel-based counterfactual estimator and shows its unbiasedness.

**Strengths:**

- The paper proposes a fundamental approach to counterfactual inference, an important topic, and can be a basis for future research in the field of counterfactual inference.
- Although the proposed approach is simple, it allows for a slight release of the stringent assumption in counterfactual inference.
- The paper is very well written and motivated.

**Weaknesses:**

- The proposed rank-preservation assumption is still very strong and unrealistic in practice.
- The mathematical notation could be introduced more thoroughly (line 87).
- The proposed method (and the loss function) only apply to continuous outcomes. However, this is not stated in the paper. Furthermore, the paper even argues with discrete outcomes (line 227).
- The related work section mostly discusses and cites works irrelevant to the topic (e.g., the paragraphs on CATE estimation and on the applications).
- The empirical evaluation is not suitable for evaluating the proposed method. It evaluates the method for estimating treatment effect (and compares it to baselines for treatment effect estimation).
However, the proposed method tackles counterfactual estimation. Although these two topics are obviously very connected, a proper evaluation of the main task of the method and a comparison with proper baselines for this task is necessary.

**Questions:**

- What is the intuition behind the definition of the introduced loss function? Could the authors please elaborate?
- How could the loss function and the method handle discrete outcomes?
- The paper states the asymptotic unbiasedness of the estimator. However, there are no statements on finite-sample performance. How does the kernel-smoothing affect the finite-sample performance (for different variable types of z)?
How does it compare to a discretized version (of the potentially continuous Z)? In the latter case, one would not need to perform smoothing.
- The method needs to learn the propensity function before the counterfactual estimation. How does the propensity estimation error propagate to the final counterfactual estimation error?
- Line 369: What is meant by "robustness of our method"?
- There are some typos in the text, and some citations are mixed up (unintentionally).

---

> ### Comment · Reviewer_vT8b · 2024-11-26
> **Answer to rebuttal**
>
> Dear authors,
>
> Thank you for addressing my questions in your rebuttal. I agree that, although restrictive, the rank-preservation assumption improves over other assumptions necessary for counterfactual estimation.
>
> However, I am not entirely convinced by all the answers presented. Specifically, my concerns regarding the following aspects are not put aside:
>
> **W3**: Thank you for the clarification. This is not clear from the manuscript. Specifically, the paper does not mention any restrictions on the domain of $Y$ in the problem setting. This should be rephrased. Without general domain restrictions, the paper is not correct in its current form.
>
> **W4**: I do not agree. The goal of the paper is counterfactual estimation, not CATE. Therefore, an in-depth literature review on CATE estimation methods is irrelevant. This directly leads to my next concerns.
>
> **W5**: Why do the authors only evaluate their method on CATE? The method is designed for counterfactual estimation. Of course, CATE estimation includes counterfactual quantities. However, this is not the only use case and, more importantly, not the goal of the paper. Therefore, in my opinion, the presentation of the method and the evaluation are not aligned. For example, it would be interesting to see how the method performs on a synthetic dataset in which both factual and counterfactual outcomes are known.
>
> **Q3  & Q4**: Regarding the derivations: Why is the weight function an arbitrary function of $X$ and $Z$? Does it need to be normalized? Furthermore, my question on the finite-sample behavior is not addressed. Asymptotically, the derivations show unbiasedness due to the consistency of the propensity estimators. However, in the real world, we are dealing with finite-sample scenarios. Are there any guarantees for this setting? Could the authors please discuss this?
>
>
> Minor notes:
>
> **W2**: In my opinion, the order of presentation is Sec. 2 is confusing for a reader unfamiliar with the notation. For example, it would help to introduce the meaning of X and Y directly in the beginning. Drawing a causal graph could also be helpful. Furthermore, it would be important to mention in the problem setting, that $Y$ has to be continuous (for the first derivations), which will later be relaxed.
>
> Overall, I would highly appreciate it if the authors could include (most of) the explanations presented in their rebuttal in an updated version of the manuscript. The primary goal of the rebuttal is to improve the paper for a better understanding of the reader, not to satisfy reviewers.

---

### Meta-Review · Area_Chair_4ncT · 2024-12-17

**Metareview:**

The paper introduces a new family of assumptions for the identifiability of the joint distribution of potential outcomes. It is a relaxation of the assumption in some recent papers. Being familiar with monotonicity in other contexts (such as $Y_x \geq Y_{x'}$ for $x \geq x'$), I found it counter-intuitive, as it relies on monotonicity on error terms, instead of more common notions of fixing $U$ while being monotonic on the causes. Basically the point is that from one single observable outcome we can infer all potential outcomes (as in the additive error model where the error is shared by all potential outcomes). I found it intriguing, but currently I don't think the material is presented clearly enough.

One major source of confusion is that Definitions 4.1 and 4.5 are claims about statistics (i.e., functions of samples, not distributions). Equation 4.2 is about a distribution. I would revise the logic of these steps with care.

Line 147 is redundant: per line 23, $Y = f_Y(X, Z, U_Y)$ (I'm not sure where ``$U_X$'' comes from). There is no need to introduce $U_x$ and $U_{x'}$. The SCM is $Y_x = f_Y(x, Z, U_Y)$. I guess that this notation is inspired by consistency equations like $Y = XY_1 + (1 - X)Y_0$ interpreted as $Y = XU_1 + (1 - X)U_0$. But in a typical SCM framing this would just be represented as the two-dimensional error term $U_Y = (U_0, U_1)$ (I understand that a common misunderstanding is the belief that error terms should be scalars - in general they can be infinite-dimensional). This non-orthodox way of using the SCM notation was already a source of confusion to me... Assumption 3.1 feels redundant given a standard SCM notation. I guess it could be interpreted as $Y$ being a non-trivial function of all of the $U_Y$ variables for all values $x$ of $X$, but then it would be unclear what being monotonic with respect to a vector. I'd phrase it instead as "the error term can be summarized by a one-dimensional quantity" (e.g., in the additive error model we assume no interaction between the latent causes and the observed causes $Y = f_1(X, Z) + f_2(U_y)$, and hence we can just define $U' = f_2(U_y)$. The same trick is used by Balke and Pearl in their original work in instrumental variables, where they use $R$ to distinguish them from $U$.)

Given that, the point is that quantiles of the conditional distribution of $Y$ can then be mapped back-and-forth between potential outcomes. This is certainly more general than the additive error model, but it doesn't mean it's more satisfying. I can make sense of additivity being believable if additivity is OKish with respect to the observed variables (generalized additive models are used in applied sciences for several reasons, one of them being not particularly bad in some domains). It can also be falsified by checking for homoscedasticity of residuals. The generality of rank preservation may actually be less appealing, since it's a vacuous claim about possible structure. Any continuous conditional distribution can be parameterized as a monotone function of a $U(0, 1)$ random variable for instance. Moving from Assumption 3.2 to 4.2 requires appreciation for even more fine-grained differences among unfalsifiable models.

It is true that even though $U_Y$ is potentially infinite-dimensional, we can fold real infinite-dimensional spaces into the real line $\mathbb R$ because they have the same cardinality, but in general we would lose the smoothness $f_Y$. That's why we can have outcomes of discrete treatments modeled with a single "error term", the structural equations not being smooth anymore. This makes the interpretation of the discrete case also harder to understand: it is lacking an interpretation of when it is sensible, and a clear contextualization of when it fails. Without these, appealing to being more general than previous papers is not sufficient. My take is that a practitioner might as well feel free to also ignore those papers. There wasn't much in the discussion or paper on trying to engage with these questions. The mainstream cross-world counterfactual constraint assumptions (e.g. $Y_x \geq Y_{x'}$ for $x \geq x'$; or additive errors) are usually argued for by proposing possible mechanisms (e.g., treatments don't harm individuals, interactions among groups of variables is weak/non-existent). The cited Xie et al. paper provides some of the insights, but e.g. for additive error models $Y = f_1(X, Z) + f_2(U)$ there is no need to start from considering something special about the functional shape of $f_2(\cdot)$, it's the notion of (lack of) interaction that is doing the heavy lifting. So even there the picture is somewhat incomplete. In this regard, it is less clear which mechanistic insight the current version of the paper is adding on top of it.

Minor:

* $Y_x = f_Y(x, z, U_x)$ is not a clear piece of notation. It seems to suggest both $x$ and $z$ being fixed. Should it be instead $Y_x = f_Y(x, Z, U_x)$ or $Y_{xz} = f_Y(x, z, U_Y)$?

**Additional Comments On Reviewer Discussion:**

Discussions were carried out in detail with the authors.

---

### Decision · Program_Chairs · 2025-01-22

Reject